# Influence-Guided Active Search for Forensic Investigation of Poisoned Training Data

## Abstract

Data poisoning attacks that inject malicious samples into training data pose a serious threat to the reliability of machine learning. Existing defense approaches focus on the fully automated detection and removal of poisoned samples; the inherent limitation of automated detection is that effective cleaning also removes a significant portion of benign samples. In contrast, we consider the *forensic investigation of poisoned data*, which relies on the verification of each sample through manual inspection, comparison with alternative data source, or some other method. The key challenge of such a forensic investigation is that the verification of each sample is expensive, but there is a limited budget for the investigation. Therefore, the investigation must strategically select, one-by-one, which samples to verify—and possibly remove—to minimize the impact of the remaining poisons. We frame this as a non-myopic sequential search problem and introduce an *influence-guided active search approach*. Our approach integrates (i) a label-free influence score that identifies training samples with disproportionate impact on test-time predictions, and (ii) an adaptive query strategy that propagates information from verified samples to focus on regions of the dataset that are both influential and likely to be poisoned. We demonstrate the efficiency and efficacy of our approach on CIFAR-10 and Tiny ImageNet against state-of-the-art attack methods, Feature Collision, Bullseye Polytope, Gradient Matching, and Narcissus. We show that our approach removes poisoned samples more effectively than fully automated cleaning methods and baseline active-search methods. This establishes our approach as a practical tool for guiding forensic investigations of poisoned training data.

## 1 Introduction

Data poisoning attacks pose a serious threat to the reliability of machine learning systems. By injecting carefully crafted malicious samples into the training data, an adversary can cause a model that is trained on the poisoned data to misclassify targeted test inputs or degrade overall performance. Clean-label poisoning attacks pose a particular challenge because they are difficult to detect: poisoned samples retain their correct labels and appear visually indistinguishable from benign ones, but they are engineered to subtly but harmfully shift decision boundaries (Fan et al., 2022). Since poisoned samples blend into the benign data, they often evade detection methods based on input anomalies, label inconsistencies, or training loss statistics (Shafahi et al., 2018; Tran et al., 2018; Jagielski et al., 2018; Turner et al., 2018).

Despite progress in defenses such as De-Pois (Chen et al., 2021), which uses model mimicry to be attack-agnostic, Deep Partition Aggregation (Levine & Feizi, 2020), which offers provable robustness, and Healthy Influential-Noise-based Training (Van et al., 2023), which reduces poison influence by adding noisy but beneficial training signals, poisoning remains difficult to detect. Most automated defenses can be bypassed by sophisticated adversarial strategies. For instance, instead of producing obvious outliers, some attacks manipulate the model into discarding innocent yet informative boundary samples, undermining the integrity of the remaining dataset (Koh et al., 2022).

Given these limitations, defending against data poisoning attacks requires more than automated data cleaning or detection. When a dataset is suspected of contamination, the defender must act as a *forensic investigator*. The goal is to verify the integrity of the training data and recover a trustworthy subset. Unlike automated cleaning, a forensic investigation is grounded in *verification*, such as

manual inspection or comparison with an alternative data source, which provides definitive evidence of whether a particular sample is poisoned or not. However, such verification steps incur a *significant cost* (e.g., manual effort from an expert), and in practice, the *budget* dedicated to an investigation (e.g., amount of effort of resource that can be spent) is limited. Therefore, a key question is how to design an investigation strategy that maximizes the utility of a limited number of verification steps, so that the limited budget yields the highest-quality dataset at the end of the investigation.

Addressing this question, however, requires more than just prioritizing which samples to verify. The defender also faces practical constraints: poisoned samples are rare and stealthy, and existing detection methods are often computationally expensive. For example, some methods rely on repeated retraining or costly gradient-Hessian calculation (Yang et al., 2022; Chhabra et al., 2024; Hammoudeh & Lowd, 2024), which are difficult at the scale of modern datasets. Thus, the challenge is twofold: the investigation must be budget-efficient while also remaining *computationally tractable*.

In our forensic setting, the defender's task is inherently sequential. Each verification step yields both the ground truth for a single sample as well as information that can guide the selection of subsequent samples to verify. This sequential nature suggests that the problem should be framed as a search process rather than as static filtering. However, existing nonmyopic active search algorithms (Jiang et al., 2017) are designed for tabular, low-dimensional data and become infeasible in high-dimensional settings. Further, these existing algorithms do not consider the potential impact of the verified samples; they simply maximize the total number of samples found by the search. The challenge, therefore, is to design a forensic strategy that (i) retains the benefits of nonmyopic active search while remaining computationally tractable for large-scale, high-dimensional poisoned data and (ii) considers the quality of the dataset remaining at the end of the investigation as its objective.

To address these challenges, we introduce a forensic formulation of active search, which is the first such formulation in the context of data poisoning to the best of our knowledge. Our setting differs from standard active search in that data samples vary significantly in importance. Some samples strongly shape the model's predictions and determine the success of an attack. Merely estimating the likelihood of a sample being poison, as the standard active search would do, overlooks this critical dimension. To address this gap, we implement an *influence-guided active search* framework for data poisoning forensics. Our method integrates two key ideas: (i) estimating the influence of training samples on model predictions to capture their potential impact, and (ii) adaptively directing queries toward regions of the data that are both highly influential and likely to contain poisons, based on verification feedback. By prioritizing samples with high *expected poison impact*, our framework efficiently allocates the scarce verification budget to maximize the discovery of poisoned data.

We demonstrate our approach on CIFAR-10 and Tiny ImageNet against three state-of-the-art poisoning attacks, Feature Collision (Shafahi et al., 2018), Bullseye Polytope (Aghakhani et al., 2021), Gradient Matching (Geiping et al., 2020), and Narcissus (Zeng et al., 2023b). Our approach differs from a fully automated cleaning method by assuming access to an expert oracle for verification. Although this introduces a *human in the loop*, it enables the investigation to retain all benign data. Existing methods for cleaning poison data, which do not consider forensic verification (e.g., EPIC (Yang et al., 2022), Meta-Sift (Zeng et al., 2023a), and PureGen (Pooladzandi et al., 2024)), must filter out or purify a significant portion of the clean samples to effectively remove the poisoned ones. By removing only those samples that are verified to be poisoned, our method neutralizes attacks by reducing Attack Success Rate to below 5%. Our method achieves this by querying as few as 0.5% of the training data to uncover poisons, while remaining computationally tractable.

The remainder of this paper is organized as follows. Section 2 reviews related work on data poisoning and cleaning and existing methods necessary for understanding our approach. Section 3 formalizes the forensic problem setting and underlying assumptions. Section 4 presents our influence-guided active search framework. Section 5 describes our experimental setup, and Section 6 presents our numerical results. Finally, Section 7 concludes the paper and discusses future directions.

## 2 RELATED WORKS AND BACKGROUND

### 2.1 RELATED WORKS

**Data Poisoning Attacks**   Data poisoning attacks compromise machine learning systems by injecting malicious samples into training datasets, disrupting model performance (Fan et al., 2022). Unlike evasion attacks, which manipulate test-phase inputs without corrupting the model itself, poisoning attacks occur during the training phase (Wang et al., 2024; Kostyumov, 2022; Biggio et al., 2013). Even a minimal injection of poisoned samples—just 0.01%—cause the model to misbehave in poisoning attacks (Carlini & Terzis, 2021). Especially, targeted attacks cause specific test instances to be misclassified into incorrect classes (Guo & Liu, 2020). Moreover, unlike label-flipping attacks (Xiao et al., 2012), clean-label attacks poison data without modifying the original labels (Peri et al., 2020; Chen et al., 2021; Goldblum et al., 2022; Tian et al., 2022; Ramirez et al., 2022) which allows long-lasting effects on the model's behavior.

**Defense Strategies against Data Poisoning**   Defense strategies against data poisoning attacks can be grouped into several main approaches. One of the most common strategies is data sanitization, which aims to identify and remove suspicious samples from the training set before training (Steinhardt et al., 2017; Koh et al., 2022). This category includes a range of methods, from unsupervised anomaly detection (Tran et al., 2018; Chen et al., 2018; Yang et al., 2022) to supervised (Zeng et al., 2023a), neighborhood-based filtering like Deep k-NN (Peri et al., 2020). While these methods can be effective when clean and poisoned data are clearly separable, they can incur significant computational costs, particularly in high-dimensional datasets like images (Qi et al., 2023).

Another line of work leverages influence functions to quantify the impact of individual training points on model behavior. This enables defenses that can filter or down-weight samples predicted to be harmful, sometimes after an initial outlier removal step (Seetharaman et al., 2022; Steinhardt et al., 2017). A related technique, HINT, perturbs influential points with "healthy noise" to reinforce beneficial patterns instead of removing them (Van et al., 2023). However, their standard formulation necessitates the approximation techniques that require a labeled test or validation set against which the influence of training data is measured.

Finally, forensic approaches offer a complementary strategy, designed to retroactively identify the source of an attack. Techniques such as iterative clustering and data unlearning can trace model misbehavior back to specific malicious training samples (Shan et al., 2022). The applicability of these methods often presumes that the specific misclassified input-output pairs that serve as a starting point for the analysis are available.

### 2.2 BACKGROUND

**Influence Scores**   Understanding the effect of individual training samples on the predictions of a complex, over-parameterized model is crucial for improving model transparency and debugging datasets. The most direct method to measure a sample's effect is to remove it from the training set, retrain the model from scratch, and measure the difference in performance. However, this leave-one-out retraining approach is computationally prohibitive for modern deep learning models.

To address this challenge, Influence Functions (Koh & Liang, 2017) were introduced as an efficient method to approximate the effect of a single training sample on a model's parameters and predictions, without the need to perform costly retraining. The core idea is to estimate the change in the model parameters, denoted as $\Delta\theta_i$, that would occur if a training sample $x_j$ were to be removed from the training data. This change in parameters can be estimated using a second-order approximation:

$$\Delta\theta_i \approx -H_\theta^{-1}\nabla_\theta\mathcal{L}(x_i, \theta) \tag{1}$$

Here, $H_\theta$ is the Hessian matrix of the total training loss with respect to the model parameters $\theta$, which captures the curvature of the loss landscape. $\mathcal{L}(x_i, \theta)$ is the loss for training sample $x_i$. The inverse Hessian $H_\theta^{-1}$ scales this gradient to approximate the complete effect on the parameters.

The traditional influence function then quantifies the influence of training sample $x_i$ with respect to a specific objective, typically the impact on the loss of a labeled test sample $x_j$. This is calculated by taking the dot product of the estimated parameter change $\Delta\theta_i$ and the gradient of the test loss for

sample $x_j$. This measures how much the loss for test sample $x_j$ would increase if sample $x_i$ were removed from the training data:

$$\mathcal{I}(x_i, x_j) = \nabla_\theta \mathcal{L}(x_j, \theta)^\top \Delta\theta_i \approx -\nabla_\theta \mathcal{L}(x_j, \theta)^\top H_\theta^{-1} \nabla_\theta \mathcal{L}(x_i, \theta) \tag{2}$$

A key limitation of this traditional approach is its reliance on a clean, labeled test dataset. The calculation of the test loss, $\mathcal{L}(x_j, \theta)$, requires the label $y_j$ of the test sample $x_j$. In a practical deployment, where the test samples are unlabeled, this method cannot be applied directly.

**Active Search** Active search is a subfield of active learning designed to identify as many members of a specific target class as possible from a large dataset, $D_{\text{train}}$, under a limited labeling budget, $B$. The process is sequential: at each step, a policy selects a single sample, $x_i$, to be evaluated by a costly oracle, returning a binary label $z_i \in \{0, 1\}$ indicating whether $x_i$ belongs to the target class (Jiang et al., 2017). The objective is to design a selection policy that maximizes the total number of target samples discovered.

To guide the search, these policies rely on a probabilistic model that, given the set of queried sample $S$, estimates the target probability $p_i$ for each unqueried sample $x_i \in U$. This is a myopic (or greedy) policy, which only considers immediate reward. At each step $t$, it queries the sample with the highest current estimated target probability:

$$x_i^* = \arg\max_{x_i \in U} \Pr(z_i = 1 | x_i, S) \tag{3}$$

While computationally efficient, this strategy does not account for how the current selection could reveal information that influences the value of subsequent queries, and thus could fail to adequately explore the search space.

Efficient Nonmyopic Search (ENS) policies (Jiang et al., 2017) accounts for the entire remaining budget, $m$, at each step. To remain computationally tractable, it approximates the expected future reward by assuming the remaining $m$ queries are made as a single, simultaneous batch. It selects the candidate $x_i$ that maximizes a score combining its immediate value with its expected impact on future rewards.

$$x_i^* = \arg\max_{x_i \in U} \left( \Pr(z_i = 1 | x_i, S) + \mathbb{E}_{z_i} \left[ \sum_{x_j \in \text{Top-}m(U \setminus \{x_i\})} \Pr(z_j = 1 | x_j, S \cup \{(x_i, z_i)\}) \right] \right) \tag{4}$$

The second term represents the expected future reward. The unknown status ($z_i$) for the candidate $x_i$ is treated as a random variable. The expectation is taken because the policy must evaluate the potential of $x_i$ before its true status is known, computing a weighted average over all possible outcomes. $S \cup \{(x_i, z_i)\}$ represents how the knowledge of $x_i$'s status updates the model and recalculates the target probabilities for the remaining samples, thus quantifying the exploratory value of the query.

## 3 PROBLEM STATEMENT

We address the data poisoning defense problem from a forensic perspective. We have a model $f_\theta$ trained on dataset $D_{\text{train}} = \{(x_i, y_i)\}_{i=1}^N$ that contains a small, unknown number of poisoned samples. The defender has access to a clean, unlabeled test set $D_{\text{test}} = \{x_j\}_{j=1}^M$ and operates under constrains: there is no clean training data, and computationally expensive methods like extensive iterative retraining cannot be afforded. The only available tool is a verification oracle that can, for a limited budget of $B$ queries, verify the true poison status $z_i \in \{0, 1\}$ of a training sample $x_i$.

### 3.1 OBJECTIVE

The goal is to obtain a filtered training set that, when used to train a model, minimizes the loss of the clean test set. This filtered dataset is constructed by querying a subset of samples $S \subseteq D_{\text{train}}$ (with $|S| \leq B$) and removing the identified poisons. The objective is therefore to select an optimal set of samples $S^*$ to be investigated, which solves the following problem:

$$S^* = \arg\min_{S \subseteq D_{\text{train}}, |S| \leq B} \mathcal{L}(f_\theta(D_{\text{train}} \setminus \{x_i \in S | z_i = 1\}), D_{\text{test}}) \tag{5}$$

However, this formulation cannot be directly solved for multiple reasons. First and foremost, we can query the verification oracle for only one set $S$, sequentially. Second, searching over all possible subsets is computationally infeasible, and each evaluation of the objective function for a given set $S$ would require retraining the model. Lastly, since we do not have labels for the test dataset, we cannot compute the actual test loss.

Given the infeasibility of directly minimizing test loss through exhaustive retraining, we introduce a computationally feasible surrogate objective. The core assumption of our forensic approach is that removing highly influential poisoned samples contributes most to reducing the final test loss. Therefore, instead of maximizing the count of identified poisons, we design a query strategy that prioritizes those with the greatest expected impact on model predictions. This reduces the problem to finding the set $S^*$ maximizes the total expected poison impact at each step:

$$S^* = \underset{S \subseteq D_{\text{train}}, |S| \leq B}{\arg\max} \sum_{x_i \in S} p_i \cdot \mathcal{I}(x_i), \tag{6}$$

where $p_i$ is the estimated probability that $x_i$ is poisoned based on oracle feedback and neighborhood consistency, and $\mathcal{I}_i(x_i)$ is the influence score of $x_i$.

## 4 INFLUENCE-GUIDED ACTIVE SEARCH

Our proposed approach adapts the standard active search described in Section 2.2 to the specific challenges of data poisoning forensics. We integrate a problem-specific reward function, expected poison impact, designed to guide the search toward data samples that are both likely to be poisoned and are maximally detrimental to the model.

### 4.1 FEATURE SPACE CONSTRUCTION

We start with training a deep neural network in the entire potentially contaminated training set $D_{\text{train}}$. The trained network, $f_\theta$, is then employed as a fixed feature extractor, mapping each training sample $x_i \in D_{\text{train}}$ to a feature vector used to construct the pairwise Euclidean distance matrix $D$. In addition, $f_\theta$ is used to calculate the influence scores, as described in the following section.

### 4.2 LABEL-FREE INFLUENCE FORMULATION

Since traditional influence functions are inapplicable in our setting without a labeled test set, we replace the test sample loss with a proposed test impact vector, $V_{\text{test}}$. We consider the model's pre-softmax output (logit vector) for each test sample and compute the average L1-norm of these vectors to quantify the overall magnitude of the predictions. The resulting $V_{\text{test}}$ represents the direction of highest sensitivity for the model's predictions on the unlabeled test set.

$$V_{\text{test}} = \nabla_\theta \left( \frac{1}{|D_{\text{test}}|} \sum_{x_j \in D_{\text{test}}} \|f_\theta(x_j)\|_1 \right) \tag{7}$$

Finally, the influence score of a training sample $x_i$ on the vulnerability of the entire unlabeled test set is calculated as follows. This is equivalent to measuring how the model parameters would shift ($\Delta\theta_i$) in the direction of $V_{\text{test}}$ if $x_i$ were removed.

$$\mathcal{I}(x_i) = (\Delta\theta_i)^\top V_{\text{test}} \approx -\left( \nabla_\theta \mathcal{L}(x_i, \theta) \right)^\top H_\theta^{-1} V_{\text{test}}. \tag{8}$$

We compute the test impact vector $V_{\text{test}}$ and the corresponding $H_\theta^{-1} V_{\text{test}}$ only once for the entire dataset. Then, for each training sample $x_i$, the influence score is calculated simply by computing its gradient $\nabla_\theta \mathcal{L}(x_i, \theta)$ and performing a single dot-product operation.

In practice, it is typically infeasible to compute the inverse Hessian matrix $H_\theta^{-1}$, so we do not compute it explicitly. Instead, we estimate the product $H_\theta^{-1} V_{\text{test}}$ using the Linear time Stochastic Second-Order Algorithm (LiSSA), which approximates the inverse Hessian-vector product through

a truncated Neumann series (Van et al., 2023; Agarwal et al., 2017). This approach avoids the prohibitive cost of computing and inverting the Hessian matrix. In our experiments (Section 5), computing this product requires approximately 2–3 minutes for a dataset. Because we have to compute it only once and we can reuse it for all training samples, this computational cost is negligible.

With $H_\theta^{-1} V_{\text{test}}$ computed, our influence formulation quantifies how each training point's gradient aligns with parameter shifts that affect the model's logit outputs. By eliminating the need for test labels, it directly captures the directional effects of poisoned samples, which are often crafted to manipulate decision boundaries in ways that simple loss-based metrics fail to reveal.

### 4.3 EXPECTED POISON IMPACT FOR SEARCH AND IMPLEMENTATION

We map each training sample to a feature vector using the extractor $f_\theta$, and we compute pairwise Euclidean distances $D$ in this feature space, as described in Section 4.1. These distances define neighborhoods that we use for estimating the probability of poisoning. At step $t$, for each unqueried sample $x_i$ (where $i \in U_t$), we estimate its probability of being poison, $p_i$. The probability is computed as the proportion of verified poisons among its $K$ nearest neighbors, where neighbors are determined using distances $D$. Letting $S_t$ denote the set of samples verified up to step $t$, the estimated probability is

$$p_i = \frac{1}{K} \sum_{j \in \mathcal{N}_i \cap S_t} z_j \tag{9}$$

where $z_j \in \{0, 1\}$ is the oracle-provided poison status for sample $x_j$.

At the outset of the search, it is possible that all verified samples are benign, i.e., $z_j = 0$ for every $j \in S_t$, which would result in $p_i = 0$ for all unqueried samples. If we select the initial $K$ samples to verify at random, then with $K = 10$ and a poison rate of 1%, the probability that the initial $K$ samples are all benign is $(1 - 0.01)^{10} \approx 90.4\%$. Even in this case, our method remains effective: as shown by the results of the ablation study in Table 3, active search without influence scores outperforms existing defenses, even though the initial sample selection is uniformly random (i.e., not guided by influence scores), demonstrating that the success of our investigation strategy does not depend on early poison discoveries. Further, in our influence-guided active search, at least one poison sample is almost always discovered among the 10 initial samples when we select the initial samples based on influence scores $\mathcal{I}(x_i)$. Once a single poison is identified, the neighborhood-based probabilities $p_i$ become informative and guide the search effectively.

The expected poison impact (EPI), $s_i$, of a sample $x_i$ is then defined as the product of its estimated poison probability and its influence score. This metric strategically directs the search toward samples that are both likely to be poisoned ($p_i$) and have strong influence on the model ($\mathcal{I}(x_i)$).

$$s_i = p_i \cdot \mathcal{I}(x_i) \tag{10}$$

We describe two versions of our framework: myopic and non-myopic. The myopic influence-guided active search instantiates the myopic search algorithm (Section 2.2) by using our expected poison impact score $s_i$ as the greedy reward function. Our non-myopic algorithm adapts the ENS framework (Jiang et al., 2017) by using the EPI score $s_i$ as the fundamental unit of reward. The total score $T_i$ for a candidate $x_i$ is calculated as its immediate reward plus the expected future reward:

$$T_i = s_i + p_i R_{\text{poison}} + (1 - p_i) R_{\text{clean}} \tag{11}$$

Here, $R_{\text{clean}}$ and $R_{\text{poison}}$ represent the total EPI expected to be collected over the remaining $m = B - t$ queries, after simulating the label for $x_i$ being in the corresponding state. Following the ENS framework, they are approximated by the sum of the top-$m$ EPI scores in the subsequent state:

$$\text{Let state} \in \{\text{poison, clean}\}. \quad R_{\text{state}} = \begin{cases} \max_{j \in U \setminus \{i\}} s'_j, & m = 1, \\ \sum \text{Top}_m(\{s'_j : j \in U \setminus \{i\}\}), & m > 1. \end{cases} \tag{12}$$

where $s'_j$ are the updated EPI scores after the simulation. To make this search tractable, ENS employs a pruning strategy based on an optimistic upper bound on future rewards. We adapt this by defining global upper bounds for the clean and poison simulations:

$$\text{UB}_{\text{clean}} = \sum \text{Top-}m(\{s_j : j \in U\}), \qquad \text{UB}_{\text{poison}} = \sum \text{Top-}m(\{s_j + \tfrac{\mathcal{I}(x_j)}{K} : j \in U\}) \tag{13}$$

The term $\frac{\mathcal{I}(x_j)}{K}$ in $\text{UB}_{\text{poison}}$ represents the maximum possible increase to a sample's EPI score if a single new poison is discovered in its neighborhood. These bounds allow the algorithm to efficiently prune candidates that cannot possibly be optimal. The complete implementation is detailed in Algorithm 1.

---

**Algorithm 1** Influence-Guided Nonmyopic Active Search

---

**Require:** Training data $D_{\text{train}}$, budget $B$, neighborhood size $K$
1: **Train Feature Extractor:** Train a DNN $f_\theta$ on $D_{\text{train}}$
2: **Compute Distances:** Use the $f_\theta$ to compute pairwise distance matrix $D \in \mathbb{R}^{N \times N}$
3: **Compute Influence Scores**: Compute the influence score vector $\mathcal{I} \in \mathbb{R}^N$
4: **Initialize Seeds:** Sort indices by descending $\mathcal{I}$; $S \leftarrow$ top-$K$ indices.
5: **if** $i \in S : z_i = 1 = \emptyset$ **then** append next highest $\mathcal{I}$ samples to $S$ until a poison is included.
6: Query oracle for $\{z_i\}_{i \in S}$. Set unqueried samples $U \leftarrow [\mathcal{N}] \setminus S$.
7: **Initialize Scores:** For each $j \in U$, find its neighbors $\mathcal{N}_j \subseteq S$. Compute $p_j \leftarrow \frac{1}{K} \sum_{i \in \mathcal{N}_j} z_i$ and
   $s_j \leftarrow p_j \cdot \mathcal{I}(x_j)$.
8: **Precompute Impact Sets:** For each $i \in U$, compute $\text{Imp}(i) \leftarrow \{j \in U : D_{ij} < \max_{l \in \mathcal{N}_j} D_{lj}\}$.
9: **for** $t = |S|$ **to** $B - 1$ **do**
10:    $m \leftarrow B - t - 1$ {Number of future steps remaining}
11:    **Compute Global Upper Bounds for Pruning:**
12:    $\text{UB}_{\text{clean}} \leftarrow \sum \text{Top-}m(\{s_j : j \in U\})$
13:    $\text{UB}_{\text{poison}} \leftarrow \sum \text{Top-}m(\{s_j + \mathcal{I}(x_j)/K : j \in U\})$
14:    **for** each candidate $i \in U$ **do**
15:       **Pruning Check:**
16:       $\text{UB}(i) \leftarrow s_i + p_i \cdot \text{UB}_{\text{poison}} + (1 - p_i) \cdot \text{UB}_{\text{clean}}$
17:       **if** $\text{UB}(i) < best\_score$ **then**
18:          **continue**
19:       **end if**
20:       **Clean Simulation** ($z_i = 0$)**:** Simulate updating scores and compute $R_{\text{clean}}$.
21:       **Poison Simulation** ($z_i = 1$)**:** Simulate updating scores and compute $R_{\text{poison}}$.
22:       $T_i \leftarrow s_i + p_i R_{\text{poison}} + (1 - p_i) R_{\text{clean}}$ {Compute total expected score}
23:       **if** $T_i > best\_score$ **or** ($T_i = best\_score$ **and** $\mathcal{I}_i > \mathcal{I}_{j^\star}$) **then**
24:          $best\_score \leftarrow T_i$; $\quad j^\star \leftarrow i$
25:       **end if**
26:    **end for**
27:    **Commit Selection:** Query oracle for label $z_{j^\star}$.
28:    **Update State:** $S \leftarrow S \cup \{j^\star\}$; $U \leftarrow U \setminus \{j^\star\}$. Update $\mathcal{N}$ and EPI for all affected points in $U$.
29: **end for**
30: **return** $S^* = \{i \in S : z_i = 1\}$

---

## 5 EXPERIMENTS

### 5.1 POISON GENERATION

We evaluate our proposed method on two standard benchmarks in adversarial machine learning and computer vision: CIFAR-10 and Tiny ImageNet. CIFAR-10 consists of 50,000 training images across 10 classes, while the Tiny ImageNet dataset contains 100,000 training images distributed among 200 classes. We generate 500 poisoned images for CIFAR-10 and 250 for Tiny ImageNet across three standard perturbation budgets ($\epsilon = 8, 16$, and $22$). The main results in Section 6 are averaged over 20 trials with distinct base-target pairs. Our poison generation process strictly follows the benchmark setup (Schwarzschild et al., 2021), adapting all original hyperparameters and optimizer configurations to ensure reproducibility.

We evaluate four data poisoning attacks: Feature Collision (FC) (Shafahi et al., 2018), Bullseye Polytope (BP) (Aghakhani et al., 2021), Gradient Matching (GM) (Geiping et al., 2020), and Narcissus (Zeng et al., 2023b). **Feature Collision (FC)** (Shafahi et al., 2018) crafts poisons designed to collide with a target's features in the model's embedding space, causing a misclassification. Similarly, **Bullseye Polytope (BP)** (Aghakhani et al., 2021) places poisons on the surface of a hypersphere centered on the target image in feature space, effectively surrounding it to enhance the attack's influence. For both the FC and BP attacks on CIFAR-10, we adapt a transfer learning setting, where a model with a pre-trained ResNet-18 is poisoned and trained for 40 epochs, starting with a

learning rate of 0.1 that decays by a factor of 10 at epochs 25 and 35. For Tiny ImageNet, poisons are generated using the BP attack ($\epsilon = 8$) with a VGG-16 model trained from scratch. **Gradient Matching (GM)** (Geiping et al., 2020) synthesizes poisons by aligning the gradients of a poisoned batch with the gradients of the target sample. As required by this method, the model is trained from scratch for 200 epochs, with a starting learning rate of 0.1 that decays by a factor of 10 at epochs 100 and 150. All three attacks are trained using SGD, with hyperparameters adapted from their original configurations. **Narcissus (Zeng et al., 2023b)** injects triggers, which are optimized using a surrogate model to represent target-class features, into a small portion of the target class training data, enabling successful attacks even with limited information about the training distribution. The attack generates poisoned samples that contain imperceptible cues, causing the model to misclassify only when the trigger is present. We replicate the Narcissus repository and follow its training setup to implement the attack, applying its backdoor-trigger generation process to the CIFAR-10 dataset under the same perturbation budgets for consistency with other attack configurations.

## 5.2 EVALUATION METRICS

We evaluate the performance of our forensic framework using two primary metrics: Attack Success Rate and test accuracy. First, **Attack Success Rate (ASR)** measures the percentage of targeted test images that the poisoned model misclassifies into the attacker's chosen class. Second, **test accuracy (Acc.)** reflects the overall performance of the trained model after the defense is applied. After removing the samples identified as poison, we measure the model's accuracy on the original clean test set. An effective defense should minimize ASR while maximizing test accuracy.

## 5.3 BASELINES

To evaluate the effectiveness of our proposed methods, we compare their performance against three state-of-the-art defense methods: Meta-Sift, EPIC, and PureGen. Note that there is a fundamental difference in strategy: whereas our approach performs a forensic investigation to precisely identify and remove only the malicious samples after an attack is suspected, these baselines operate as data cleaning mechanisms. Their goal is to sanitize (or purify) the training set by removing (or reconstructing) a subset of suspicious data samples, which can inadvertently include benign samples.

**Meta-Sift** (Zeng et al., 2023a) is a defense mechanism that empowers users to specify the number of images to sift from the training process. For our experiments, we configured Meta-Sift to sift out a thousand images from the dataset. This parameter was selected by considering the size of the dataset and the number of poisoned images included in the dataset. **EPIC** (Yang et al., 2022) is another defense strategy against data poisoning attacks. It uses a clustering approach to detect examples that are not close to other examples in the gradient space and omits these isolated examples during training. In this way, it can prevent the misclassifications of the target image at inference time. **PureGen** (Pooladzandi et al., 2024) is a defense method that uses the stochastic dynamics of energy-based models or diffusion models to purify the training dataset by reconstructing samples to eliminate adversarial perturbations.

# 6 RESULTS AND ANALYSIS

## 6.1 OVERALL PERFORMANCE

The results demonstrate the effectiveness of our proposed forensic approach. As shown in Table 1 and Table 2, our method consistently outperforms in reducing the Attack Success Rate (ASR) while maintaining or improving test accuracy across different datasets, attack types, and perturbation sizes.

On the CIFAR-10 dataset, we evaluate our method, Influence-Guided Active Search, with a query budget of $B = 250$, which is only 0.5% of the 50,000 sample training set. This small budget proves highly effective. For instance, against the potent BP attack, which achieves an ASR of up to 95% on the undefended model, our approach neutralizes its effectiveness, reducing the ASR to 0% for $\epsilon = 8$ and $\epsilon = 16$, and to just 5% for $\epsilon = 22$. While EPIC also reduces ASR, our method is often more effective, as demonstrated in the GM attack scenarios, where our method achieves lower ASR at higher epsilon values. Moreover, against the Narcissus backdoor attack, our method achieves the lowest ASR and the highest test accuracy, outperforming both Meta-Sift and EPIC.

A distinct advantage of our approach is its ability to preserve or even enhance test accuracy. Across all experiments, models defended by our method achieve higher accuracy than both the undefended models and those protected by competing data-cleaning methods. This is because our precision-oriented forensic technique selectively removes only the identified poisoned samples. In contrast, other defenses like Meta-Sift and EPIC often show smaller gains or even a decline in accuracy. Their data-cleaning mechanisms disregards a substantial number of benign samples along with the poisons, thereby degrading the training data quality and harming overall model performance.

Table 1: Attack Success Rate (ASR) and average test accuracy (Acc.) over 20 trials on the CIFAR-10 dataset poisoned with three perturbation sizes.

| | Undefended | | Influence-Guided Active Search | | Meta-Sift | | EPIC | | PureGen | |
|---|---|---|---|---|---|---|---|---|---|---|
| | ASR ($\downarrow$) | Acc. ($\uparrow$) | ASR ($\downarrow$) | Acc. ($\uparrow$) | ASR ($\downarrow$) | Acc. ($\uparrow$) | ASR ($\downarrow$) | Acc. ($\uparrow$) | ASR ($\downarrow$) | Acc. ($\uparrow$) |
| $\epsilon = 8$ | | | | | | | | | | |
| FC | 10% | 0.9460 | 0% | 0.9497 | 10% | 0.9423 | 0% | 0.9467 | experiments still running... | |
| BP | 90% | 0.9457 | 0% | 0.9500 | 20% | 0.9455 | 5% | 0.9467 | ... | |
| GM | 25% | 0.9487 | 5% | 0.9486 | 20% | 0.9407 | 5% | 0.8899 | ... | |
| $\epsilon = 16$ | | | | | | | | | | |
| FC | 15% | 0.9456 | 0% | 0.9468 | 15% | 0.9325 | 0% | 0.9466 | ... | |
| BP | 95% | 0.9458 | 0% | 0.9475 | 35% | 0.9342 | 5% | 0.9465 | ... | |
| GM | 55% | 0.9486 | 15% | 0.9476 | 40% | 0.9352 | 30% | 0.8900 | ... | |
| Narcissus | 91.20% | 0.9521 | 7.09% | 0.9551 | 45.54% | 0.9526 | 11.95% | 0.7360 | ... | |
| $\epsilon = 22$ | | | | | | | | | | |
| FC | 20% | 0.9455 | 0% | 0.9483 | 15% | 0.9456 | 0% | 0.9466 | ... | |
| BP | 95% | 0.9458 | 5% | 0.9503 | 55% | 0.9386 | 5% | 0.9468 | ... | |
| GM | 20% | 0.9491 | 5% | 0.9465 | 20% | 0.9373 | 20% | 0.8901 | ... | |

We further validate our method's scalability on Tiny ImageNet. We again set the query budget to $B = 250$, which matches the number of poisoned samples in the training set. As shown in Table 2, Influence-Guided Active Search again outperforms Meta-Sift and EPIC in terms of both ASR and accuracy, reducing ASR from 30% to 15% while improving accuracy from 0.6153 to 0.6218.

Table 2: Attack Success Rate (ASR) and average test accuracy (Acc.) over 20 trials on the Tiny ImageNet dataset poisoned by BP attack with the perturbation size of 8.

| Undefended | | Influence-Guided Active Search | | Meta-Sift | | EPIC | | PureGen | |
|---|---|---|---|---|---|---|---|---|---|
| ASR ($\downarrow$) | Acc. ($\uparrow$) | ASR ($\downarrow$) | Acc. ($\uparrow$) | ASR ($\downarrow$) | Acc. ($\uparrow$) | ASR | Acc. ($\uparrow$) | ASR ($\downarrow$) | Acc. ($\uparrow$) |
| 30% | 0.6153 | 15% | 0.6218 | 25% | 0.6011 | 25% | 0.5699 | ... | |

## 6.2 ABLATION STUDY

To isolate the contributions of our key components—influence-guided search and non-myopic lookahead—we perform an ablation study. We evaluate four variants of our method: **Myopic**, a greedy search without influence scores; **Myopic-Influence**, a greedy search guided by influence scores; **Two-Step**, a search that looks ahead one more step without influence scores; and **Two-Step-Influence**, a two-step lookahead search guided by influence scores. Unlike the full non-myopic method, which simulates outcomes over the entire remaining budget, the "Two-Step" variant provides a lightweight approximation by looking ahead one more step. For each candidate, it simulates both possible oracle outcomes (poison or clean) and selects the action that leads to the best achievable score in the subsequent step. Table 3 reports the results of this study on the CIFAR-10 and Tiny ImageNet datasets. The findings validate the importance of our core components.

First, incorporating influence scores consistently improves performance. Comparing the base methods to their "-Influence" counterparts (e.g., Myopic vs. Myopic-Influence) shows that influence guidance generally reduces the ASR or improves test accuracy. In the CIFAR-10 GM attack ($\epsilon = 16$), adding influence scores lowers the ASR from 15% to 10% without sacrificing accuracy in both the Myopic and Two-Step settings. Second, while the limited lookahead provides some benefit, its impact is less pronounced. The Two-Step search offers marginal improvement over the purely greedy Myopic search in specific scenarios, like the BP attack on Tiny ImageNet (ASR drops from 20% to 15%). However, these simplified variants remain less effective than our full Nonmyopic-

Influence approach, confirming that the combination of influence guidance and a comprehensive lookahead strategy is crucial for achieving optimal performance.

Table 3: Attack Success Rate (ASR) and average test accuracy (Acc.) over 20 trials on two different datasets poisoned with three perturbation sizes. Myopic search with and without influence scores as well as Two-Step search with and without influence scores are evaluated. The budget is set to 250.

| | Myopic | | Myopic-Influence | | Two-Step | | Two-Step-Influence | | Influence-Guided Active Search | |
|---|---|---|---|---|---|---|---|---|---|---|
| | ASR ($\downarrow$) | Acc. ($\uparrow$) | ASR ($\downarrow$) | Acc. ($\uparrow$) | ASR ($\downarrow$) | Acc. ($\uparrow$) | ASR ($\downarrow$) | Acc. ($\uparrow$) | ASR ($\downarrow$) | Acc. ($\uparrow$) |
| **CIFAR-10** | | | | | | | | | | |
| *$\epsilon = 8$* | | | | | | | | | | |
| FC | 0% | 0.9489 | 0% | 0.9482 | 0% | 0.9473 | 0% | 0.9492 | 0% | 0.9497 |
| BP | 5% | 0.9465 | 5% | 0.9476 | 5% | 0.9483 | 5% | 0.9495 | 0% | 0.9500 |
| GM | 10% | 0.9401 | 5% | 0.9479 | 10% | 0.9466 | 10% | 0.9466 | 5% | 0.9486 |
| *$\epsilon = 16$* | | | | | | | | | | |
| FC | 5% | 0.9467 | 0% | 0.9479 | 0% | 0.9482 | 0% | 0.9484 | 0% | 0.9468 |
| BP | 0% | 0.9486 | 0% | 0.9507 | 0% | 0.9484 | 0% | 0.9493 | 0% | 0.9475 |
| GM | 15% | 0.9342 | 10% | 0.9386 | 15% | 0.9379 | 10% | 0.9379 | 15% | 0.9476 |
| *$\epsilon = 22$* | | | | | | | | | | |
| FC | 0% | 0.9461 | 0% | 0.9474 | 5% | 0.9478 | 5% | 0.9485 | 0% | 0.9483 |
| BP | 5% | 0.9465 | 0% | 0.9477 | 5% | 0.9475 | 0% | 0.9481 | 5% | 0.9503 |
| GM | 5% | 0.9350 | 5% | 0.9363 | 10% | 0.9349 | 10% | 0.9396 | 5% | 0.9465 |
| **Tiny Imagenet** | | | | | | | | | | |
| BP ($\epsilon = 8$) | 20% | 0.6176 | 20% | 0.6194 | 15% | 0.6200 | 20% | 0.6161 | 15% | 0.6218 |

## 6.3 LABEL-FREE INFLUENCE SCORE

To demonstrate that our label-free influence metric (Eq. (8)) is a good surrogate for the true objective of labeled test influence (Eq. (6)), we measure the correlation between these two metrics over poisoned samples. We also compare our proposed influence metric, which is the L1 norm of the logit values output by the model, to two alternative metrics: *L2 norm of the logit values* and *entropy of the class distribution output by the model*. Each metric provides a label-free quantification of the impact that a sample may have on the test loss, and therefore, they could all serve as practical surrogates for the true objective of labeled test influence.

In this comparison, we consider the CIFAR-10 dataset poisoned using FC, BP, and GM attacks with $\epsilon = 16$. For each attack and each metric, we compute the Spearman correlation between the metric and the true objective over all poisoned samples. Table 4 shows the mean correlation ($\pm$ its standard deviation) measured over 20 poisoned datasets in each case. The results show that our proposed, L1-norm based influence metric achieves the highest correlation with the true objective of labeled test influence, and it does so with lower or comparable variance as the L2-norm and entropy metrics. This indicates that our proposed metric provides a stable and reliable approximation of the true objective, making it the best choice in practice for the forensic investigation of poisoned datasets.

Table 4: Correlation between each proxy metric and the labeled test influence of poisoned samples, aggregated over 20 poisoned datasets (mean $\pm$ std.).

| Attack Types | L1 Norm | L2 Norm | Entropy |
|---|---|---|---|
| FC | **0.39 $\pm$ 0.08** | 0.33 $\pm$ 0.13 | 0.18 $\pm$ 0.14 |
| BP | **0.42 $\pm$ 0.10** | 0.37 $\pm$ 0.05 | 0.06 $\pm$ 0.08 |
| GM | **0.22 $\pm$ 0.25** | 0.22 $\pm$ 0.26 | 0.09 $\pm$ 0.27 |

## 7 CONCLUSION

This work introduces an influence-guided active search framework to defend against data poisoning in a forensic setting where oracle queries are limited. Our method strategically combines influence scores and poisoning probability estimates to efficiently identify malicious data. Experiments on CIFAR-10 and Tiny ImageNet show our framework effectively neutralizes state-of-the-art poisoning attacks, significantly reducing their success rates while maintaining or improving test accuracy. The results confirm that treating data cleaning as an active forensic investigation yields superior model robustness and utility compared to traditional defenses. Future work will focus on scaling the framework to larger datasets and broader threat models.

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

## A  The Use of Large Language Models (LLMs)

We acknowledge the use of large language models to support the writing process, including grammar correction and refining the flow of sentences and paragraphs.

## B  Notations

Table 5 summarizes the notations used throughout this paper. We distinguish between training and test datasets, poisoned and clean samples, as well as the key variables employed in our forensic investigation framework. The notations cover dataset indices, query budget, neighbor sets, influence scores, and probability estimates. They are essential to formally define our problem setup and proposed method. For completeness, we also include $\epsilon$, which denotes the magnitude of the poisoning perturbation used in our experiments. This notation table serves as a reference to ensure consistency and clarity across the main text and supplementary materials.

Table 5: Overview of Notations

| Notation | Description |
|---|---|
| **General Notations** | |
| $D_{\text{train}}$ | The training dataset |
| $N$ | Total number of samples in $D_{\text{train}}$ |
| $x_i$ | The $i$-th training sample |
| $y_i$ | Ground-truth label of the training sample $x_i$ |
| $z_i$ | Binary status for sample $x_i$ (1: poisoned/target, 0: clean/non-target) |
| **Active Search Notations** | |
| $B$ | Total query budget for the active search |
| $t$ | Current step/iteration in the active search $(0, 1, \ldots, B-1)$ |
| $m$ | Number of remaining queries at step $t$, $m = B - t$ |
| $S_t$ | Set of samples selected (queried) up to step $t$ |
| $U_t$ | Set of unselected (unqueried) samples at step $t$, $U_t = D_{\text{train}} \setminus S_t$ |
| $x_t^*$ | The sample selected by the policy at step $t$ |
| $p_i$ | Estimated poison probability for sample $x_i$, $p_i = \Pr(z_i = 1 | x_i, S_t)$ |
| **Method-Specific Notations** | |
| $D$ | Pairwise distance matrix $(N \times N)$ for the training set |
| $K$ | Number of nearest neighbors for probability estimation |
| $\mathcal{N}_i$ | Set of $K$-nearest neighbors of sample $x_i$ |
| $\mathcal{I}$ | Vector of influence scores for all training samples |
| $\epsilon$ | Size of the poisoning perturbation |
| $D_{\text{test}}$ | The test dataset |

## C  Active-Search Hyperparameter $K$

The active-search hyperparameter $K$ determines both (1) the number of verified neighbors that we consider when estimating the poison probability $p_i$ of an unqueried sample $x_i$ and (2) the number of initial samples, which are chosen at the beginning of the search based only on their influence scores $\mathcal{I}(x_i)$, before we have enough verified samples to estimate probabilities. Choosing an appropriate value for $K$ has to consider not only the effectiveness of the search, that is, what value of $K$ maximizes the number of poisoned samples found by the search with a given investigation budget $B$; it should also consider computational cost since higher values of $K$ lead to increased running time for the active-search algorithm.

To choose a practical value for $K$, we conducted a hyperparameter search over a range of candidate values: $K \in \{1, 3, 5, 7, 10, 15, 20, 30\}$. Here, for each candidate value of $K$, we present the results

Table 6: Average number of poisoned samples discovered by active search, for various values of $K$, in datasets poisoned using Gradient Matching (GM) attack.

| $K$ | $\epsilon = 8$ | $\epsilon = 16$ | $\epsilon = 22$ |
|---|---|---|---|
| 1 | 75.3 | 59.7 | 144.0 |
| 3 | 84.3 | 68.7 | 155.3 |
| 5 | 86.3 | 69.3 | 160.0 |
| 7 | 88.3 | 72.0 | 158.3 |
| **10** | **100.0** | **80.7** | **168.3** |
| 15 | 89.7 | 75.0 | 148.0 |
| 20 | 88.0 | 72.7 | 148.0 |
| 30 | 87.7 | 71.3 | 134.0 |

of running influence-guided, non-myopic active search on datasets poisoned by the Gradient Matching (GM) attack with perturbation sizes $\epsilon = 8$, 16, and 22. In each case, we evaluated the search on three poisoned datasets and calculated the average number of poisons discovered on these datasets.

Table 6 shows that the number of discovered poisoned samples increases as $K$ increases to $K = 10$; beyond this, the number of discovered poisons begins to gradually decrease as we further increase $K$. Moreover, the search does not appear to be particularly sensitive to hyperparameter $K$ since the changes in the number of discovered poisons are relatively modest as we vary the value of $K$ around $K = 10$. We adopt $K = 10$ in all experiments because it provides a stable balance: it generally leads to the highest number of discovered poisoned samples, avoiding neighborhood dilution that occurs when too many neighbors are aggregated, and it keeps the computational cost of the search relatively low.

