# OpenReview forum: "Influence-Guided Active Search for Poisoned Data Forensics"
_ICLR.cc/2026/Conference — Submitted to ICLR 2026_

### Official Review · Reviewer_yoJN · 2025-10-31

**Soundness:** 2
**Presentation:** 3
**Contribution:** 3
**Rating:** 6
**Confidence:** 3

**Summary:**

This paper suggests that fully automated detection and removal of poisoned samples has an inherent limitation in also removing a significant portion of benign samples. In response, this paper proposes a novel framework that recasts the problem as a "forensic investigation," where a human expert can verify a limited number of samples. The core contribution is an influence-guided active search algorithm designed to maximize the impact of this limited verification budget. The method sequentially selects samples for verification by prioritizing those with the highest expected poison impact.

**Strengths:**

- Moving from fully-automated data cleaning to a budget-constrained forensic investigation is a practical and highly relevant shift.

- The proposed EPI metric is an intuitive solution. The ideal sample to check should be both likely to be poisoned and highly influential on the model behavior sounds reasonable.

- I think the label-free influence score is significant. It overcomes a major limitation of standard influence functions, which require a labeled test or validation set, making the proposed method applicable to more realistic deployment settings.

**Weaknesses:**

- The proposed method relies on a Hessian-based influence function. While this term is computed only once, it might still be computationally expensive for the large-scale models common today. A more detailed discussion of the practical computational cost would be beneficial.

- The poison probability is estimated using K-NN in the feature space of the poisoned model. This assumes that poisoned samples are somewhat clustered or near each other. If an attack is extremely stealthy and its samples are well-scattered, the K-NN estimate might be unreliable, weakening the EPI metric.

- The proposed method queries the top-K most influential samples until a poison is found. If the most influential samples are all benign, the search may be slow to start or even fail if the budget is exhausted before finding the first poison. The paper would benefit from a discussion on the sensitivity to this.

**Questions:**

- The poison probability estimate depends on the neighborhood size, K. How sensitive is the performance of the proposed method to this hyperparameter? How was K selected for the experiments?

- How does the proposed method perform if the initial top-K influential samples are all benign? Have you explored alternative strategies to find the first poison, and how do they compare?

- The paper focuses on clean-label, targeted attacks. How do you expect this framework to perform against other poisoning modalities, such as untargeted availability attacks (which just aim to degrade overall accuracy)? Would the influence score still be the correct heuristic?

---

> ### Author Response · Authors · 2025-11-23
>
> ### Q1. Hyperparameter Sensitivity and Search
>
> We agree with the reviewer that it is important to study how sensitive our proposed approach is to hyperparameter $K$ and what the optimal value is for $K$. This hyperparameter controls both (1) the number of verified neighbors that we consider when estimating the poison probability $p_i$ of an unqueried sample $x_i$ and (2) the number of initial samples, which are chosen at the beginning of the search based only on their influence scores $\mathcal{I}(x_i)$, before we have enough verified samples to estimate probabilities.
>
> To study sensitivity and choose an optimal value for $K$, we conducted a hyperparameter search over a range of candidate values: $K \in \{1,3,5,7,10,15,20,30\}$. We have included these new results in the paper as Table 6 (see Appendix C in the revised paper PDF) as well as in the table below. For each candidate value of $K$, we applied our proposed approach on datasets poisoned by the Gradient Matching (GM) attack with perturbation sizes $\epsilon$ = 8, 16, and 22. In each case, we evaluated the search on three poisoned datasets and calculated the average number of poisons discovered over these datasets.
>
> | $K$ | $\epsilon$ =  8 | $\epsilon$ = 16 | $\epsilon$ = 22 |
> |-----|--------|--------|--------|
> |   1 |   75.3 |   59.7 |  144.0 |
> |   3 |   84.3 |   68.7 |  155.3 |
> |   5 |   86.3 |   69.3 |  160.0 |
> |   7 |   88.3 |   72.0 |  158.3 |
> |  **10** |  **100.0** |   **80.7** |  **168.3** |
> |  15 |   89.7 |   75.0 |  148.0 |
> |  20 |   88.0 |   72.7 |  148.0 |
> |  30 |   87.7 |   71.3 |  134.0 |
>
> The results show that the number of discovered poisoned samples increases as $K$ increases to $K=10$; beyond this, the number of discovered poisons begins to gradually decrease as we further increase $K$. Moreover, the search does not appear to be particularly sensitive to hyperparameter $K$ since the changes in the number of discovered poisons are relatively modest as we vary the value of $K$ around $10$. We adopt $K=10$ in all experiments because it provides a stable balance: it generally leads to the highest number of discovered poisoned samples, avoiding neighborhood dilution that occurs when too many neighbors are aggregated, and it keeps the computational cost of the search relatively low.

---

> > ### Author Response · Authors · 2025-11-23
> >
> > ### Q2 / W3. Only Clean Samples in Initial Selection
> >
> > We thank the reviewer for pointing out the case where all initially queried samples are clean (i.e., benign); in this case, the estimated poison probability is $p_i = 0$ for every  unqueried sample $x_i$ at the beginning of the search. This situation is common under low poison rates but does not degrade search performance. With a 1% of poison rate and $K=10$ initial queries, the probability that all queried samples are clean is $(1-0.01)^{10} \approx 90.4$% if the initial samples are selected uniformly at random. However, even in this case, our method remains effective as shown by the ablation results in Table 3 (in Section 6.2). These results show that active search without influence scores, which selects initial queries uniformly at random, already outperforms the baseline defenses. This confirms that the success of our search strategy does not depend on early poison discoveries.
> >
> > Moreover, in our influence-guided active search method, we almost always discover at least one poisoned sample within the first $K=10$ queries, because the influence score reliably ranks poisoned samples near the top. Once a single poisoned sample is discovered, the neighborhood-based probabilities $p_i$ become informative and the search improves rapidly. Thus, the “all-clean initial set” scenario neither breaks nor meaningfully degrades our approach in practice.

---

> > > ### Author Response · Authors · 2025-11-23
> > >
> > > ### Q3 / W2. Performance on a Different Threat Models
> > >
> > > We appreciate the reviewer’s question regarding the assumptions and the suggestion to broaden our evaluation. We would like to clarify that *our proposed approach does not assume that the poisoning attack is clean label*. We focused on Feature Collision (FC), Bullseye Polytope (BP), and Gradient Matching (GM) in our evaluation because these are representative data-poisoning attacks, which are widely used in the literature to benchmark defense methods.
> > >
> > > While it is true that the *K*-NN-based probability estimate depends on locality in the feature space, poisoning attacks generally rely on sparse and locally consistent perturbations to remain inconspicuous. Even when this structure is weak, our Expected Poison Impact metric can remain effective. When poisoned samples are more scattered, the influence score naturally becomes the dominant signal, since any impactful poison must exert disproportionate pressure on the model to alter decision boundaries.
> > >
> > > Following the reviewer's suggestion, we have broadened the threat models considered in our experimental evaluation by including a new backdoor poisoning attack. We chose **Narcissus**, a *state-of-the-art backdoor attack* (Zeng et al., 2023b). We evaluated our proposed approach and both baseline methods, Meta-Sift and EPIC, on the CIFAR-10 dataset poisoned by Narcissus. We have included these new results in Table 1 (see revised paper PDF) as well as in the table below.
> > >
> > > |Narcissus ($\epsilon = 16$)|Undefended|Influence-Guided Active Search|Meta-Sift|EPIC|
> > > | --------------------------- | ---------- | ------------------------------ | ---- | --------- |
> > > |Attack Success Rate (↓) | 91.20% | **7.09%** | 45.54% | 11.95% |
> > > |Test Accuracy (↑) | 0.9521 | **0.9551** | 0.9526 | 0.7360 |
> > >
> > > The results demonstrate that our proposed approach is *effective against backdoor poisoning* since it achieves significantly lower ASR than the cleaning methods, while also achieving higher test accuracy. Notably, the results very clearly illustrate the advantages of a forensic investigation: automated cleaning methods face a trade-off between suffering high ASR (Meta-Sift) or low test accuracy (EPIC).
> > >
> > > We are currently running experiments with Narcissus on the *Tiny ImageNet* dataset. We will include the results in the paper as soon as they are ready.

---

> > > > ### Author Response · Authors · 2025-11-23
> > > >
> > > > ### W1. Computational Cost of Hessian-based Influence Scores
> > > >
> > > > The reviewer raises a very good point. Indeed, explicitly computing the inverse Hessian matrix $H_{\theta}^{-1}$ is typically infeasible in practice due to the sheer size (i.e., $|\theta|\times|\theta|$) of matrix $H_{\theta}$. We have revised Section 4.2 (see below Equation (8) in the revised paper PDF) to clarify that we do not compute the inverse Hessian matrix itself. Instead, we directly estimate the product $H_{\theta}^{-1} \cdot V_{test}$ following a similar approach as prior works (Van et al., 2023; Agarwal et al., 2017). In our experiments, computing this product takes only 2-3 minutes for a given dataset. Since we have to compute it only once and we can reuse it for all training samples, this computational cost is negligible relative to other costs, such as model training.

---

### Official Review · Reviewer_AY6X · 2025-10-31

**Soundness:** 2
**Presentation:** 2
**Contribution:** 2
**Rating:** 2
**Confidence:** 2

**Summary:**

This paper introduces Influence-Guided Active Search (IGAS), a forensic approach to defend machine learning models against data poisoning attacks. Instead of relying on automated data cleaning, IGAS strategically selects samples for verification under a limited budget. It combines a label-free influence score, which measures each sample’s impact on model predictions without using labeled test data, with an adaptive active search strategy that focuses on samples most likely to be poisoned and influential. Experiments on CIFAR-10 and Tiny ImageNet show that IGAS outperforms existing defenses such as Meta-Sift and EPIC, effectively neutralizing attacks while preserving model accuracy using less than 0.5% of the training data.

**Strengths:**

The paper introduces a well-motivated shift from automated defenses to a forensic, human-in-the-loop approach for data poisoning mitigation. Its main strength lies in integrating influence analysis with active search, enabling efficient and targeted verification under limited resources. The proposed label-free influence formulation is practical, removing the need for labeled test data while effectively capturing each sample’s impact. Experiments are comprehensive across multiple datasets and attack types, showing strong empirical performance by reducing attack success rates with minimal queries. Overall, the method is conceptually novel, computationally efficient, and empirically convincing, offering a solid foundation for future forensic approaches in data security.

**Weaknesses:**

1.The method assumes access to a reliable oracle for sample verification, which may not be feasible or scalable in real-world applications. It also presumes that poisoned samples are sparse and locally consistent, an assumption that might fail under adaptive or distributed poisoning attacks.
2.The paper only compares with Meta-Sift and EPIC, missing stronger baselines such as Towards a Proactive ML Approach for Detecting Backdoor Poison Samples (Qi et al., 2023), and Multidomain active defense: Detecting multidomain backdoor poisoned samples via ALL-to-ALL decoupling training without clean datasets (MAD, 2024). These recent methods address sample poisoning more proactively and in multi-domain settings without relying on clean data.
3. The proposed method focuses primarily on clean-label poisoning attacks, such as Feature Collision, Bullseye Polytope, and Gradient Matching, and does not address trigger-based backdoor attacks.

**Questions:**

1. **Scope and Applicability**
   The paper focuses on clean-label poisoning attacks. How might the proposed influence-guided framework perform under dirty-label or backdoor poisoning scenarios?

2. **Comparative Evaluation**
   The study compares mainly against Meta-Sift and EPIC. How would IGAS perform relative to more recent defenses like Proactive ML Detection, or Multidomain Active Defense?

3. **Assumption Robustness**
   The approach assumes that poisoned samples are rare and locally clustered. What are the potential weaknesses if this assumption fails, such as in adaptive or distributed attacks?

---

> ### Author Response · Authors · 2025-11-23
>
> ### Q1 / W3. Backdoor Poisoning Attacks
>
> We thank the reviewer for the suggestion to consider other types of attacks, such as trigger-based backdoor attacks. We would like to clarify that *our proposed approach does not assume that the poisoning attack is clean label*. We focused on Feature Collision (FC), Bullseye Polytope (BP), and Gradient Matching (GM) in our evaluation because these are representative data-poisoning attacks, which are widely used in the literature to benchmark defense methods.
>
> Following the reviewer's suggestion, we have extended our experimental evaluation by including a new attack, **Narcissus**, a *state-of-the-art backdoor poisoning attack* (Zeng et al., 2023b). We have evaluated our proposed approach and both baseline methods, Meta-Sift and EPIC, on the CIFAR-10 dataset poisoned by Narcissus. We have included these new results in Table 1 (see revised paper PDF) as well as in the table below.
>
> |Narcissus ($\epsilon = 16$)|Undefended|Influence-Guided Active Search|Meta-Sift|EPIC|
> | --------------------------- | ---------- | ------------------------------ | ---- | --------- |
> |Attack Success Rate (↓) | 91.20% | **7.09%** | 45.54% | 11.95% |
> |Test Accuracy (↑) | 0.9521 | **0.9551** | 0.9526 | 0.7360 |
>
> The results demonstrate that our proposed approach is *effective against backdoor poisoning* since it achieves significantly lower ASR than the cleaning methods, while also achieving higher test accuracy. Notably, the results very clearly illustrate the advantages of a forensic investigation: automated cleaning methods face a trade-off between suffering high ASR (Meta-Sift) or low test accuracy (EPIC).
>
> We are currently running experiments with Narcissus on the *Tiny ImageNet* dataset. We will include the results in the paper as soon as they are ready.

---

> > ### Author Response · Authors · 2025-11-23
> >
> > ### Q2 / W2. Comparative Evaluation
> >
> > We thank the reviewer for the helpful suggestions; the two papers suggested by the reviewer (Qi et al., 2023 and Ma et al., 2023) are indeed highly relevant to our work.
> >
> > To strengthen our experimental evaluation, we have included a new defense baseline, **PureGen**, a *state-of-the-art data cleaning method* (Pooladzandi et al., 2024). We chose PureGen due to its recency (published at NeurIPS 2024) and its state-of-the-art performance. We have started running experiments, which compare our proposed approach to PureGen on datasets poisoned by each attack (FC, BP, GM, and Narcissus); however, these experiments have not finished yet. We will update the paper (revised PDF) as well as this comment with the results as soon as they are ready.

---

> > > ### Author Response · Authors · 2025-11-23
> > >
> > > ### Q3 / W1. Robustness of Assumptions: Verification Oracle and Poison Sparsity
> > >
> > > We appreciate the thoughtful question. We would like to clarify that our method is designed for a forensic scenario, where the oracle represents a human expert or a rigorous verification process. In safety-critical domains, such as autonomous systems and medical diagnosis, relying on automated filtering may be insufficient because automated methods tend to have high false-negative rates. Therefore, expert intervention may be necessary to ensure accountability and model safety. We agree that forensic verification is a limited and expensive resource. This is exactly why our framework considers a limited verification budget and focuses on optimizing the selection of samples to be verified. By formulating the problem as an active search, our approach prioritizes investigating samples that are suspicious, may have high impact on test outputs, and can inform subsequent steps of the search, thereby maximizing the total influence of poisoned samples discovered with a limited budget. This ensures that each oracle query provides maximal benefit when verification resources are scarce.
> > >
> > > Regarding the assumptions of poison sparsity and consistency, we do not explicitly assume that poisoned samples are sparse or locally consistent in our problem formulation or in our experiments. Instead, we focus on state-of-the-art poisoning attacks, which often rely on sparse and locally consistent perturbations to remain undetected but effective. Therefore, sparsity emerges naturally as a defining constraint for strong, undetectable poisoning attacks. Even for a backdoor attack, our approach of prioritizing samples with the highest expected poison impact is efficient, as demonstrated by our experimental results on Narcissus.

---

### Official Review · Reviewer_VVQc · 2025-11-02

**Soundness:** 3
**Presentation:** 3
**Contribution:** 3
**Rating:** 6
**Confidence:** 3

**Summary:**

This paper reframes data-poisoning defense as budgeted forensic verification, where a human oracle inspects selected training samples rather than relying on fully automated detection. The central challenge is to maximize utility under a small verification budget. The authors propose an influence-guided active search that (i) computes a label-free influence score estimating how each training sample perturbs model predictions on an unlabeled test set, and (ii) applies an adaptive query policy that prioritizes samples with the highest expected impact. Across CIFAR-10 and Tiny ImageNet under Feature Collision, Bullseye Polytope, and Gradient Matching attacks, the method reduces attack success rates to <5% while maintaining test accuracy, demonstrating effective, budget-aware poisoning forensics.

**Strengths:**

1) **Novel problem framing.**
   Recasting poisoning defense as a **forensic, human-in-the-loop verification** problem—rather than a fully automated detection task—is both practical and distinctive, aligning with real workflows where limited expert review is budgeted for maximal impact.

2) **Label-free influence formulation.**
   The adaptation of influence functions (Eqs. 7–8) to operate **without labeled test data**—using the **L1 norm of model logits** as a proxy—addresses a common deployment constraint and makes the influence signal broadly usable.

3) **Strong empirical results.**
   The method delivers **large ASR reductions** (often from **20/20 to ≤1/20** cases) while **maintaining or improving test accuracy**, outperformin

**Weaknesses:**

1) **Limited technical novelty.** The method largely adapts existing components—an ENS-style active search (Jiang et al., 2017) and standard influence functions (Koh & Liang, 2017) with a straightforward unlabeled-data variant—and combines them via a simple expected-impact objective \(p_i \cdot I_i\). While the **label-free influence** is the most distinctive element, it currently lacks a theoretical justification for why the **L1 norm of logits** captures vulnerability, an empirical check against **labeled** influence as a sanity baseline, and a rationale for choosing **L1** over alternatives (e.g., **L2**, margin/entropy). Overall, the contribution reads as a pragmatic **reformulation and integration** rather than a fundamentally new algorithmic advance.

2) **Heuristic influence proxy.**
   The L1-logit “test-impact” proxy is plausible but ad-hoc; provide analysis of **when it correlates or fails to correlate** with true test loss to justify its use and guide practitioners.

3) **Limited evaluation breadth.**
   Current results are on **CIFAR-10/Tiny-ImageNet** with three attacks. To improve external validity, add **broader threat models** (e.g., backdoors, multi-target poisons) and **larger-scale datasets**

**Questions:**

Please refer to the Weaknesses

---

> ### Author Response · Authors · 2025-11-23
>
> ### W1. Technical Novelty
>
> We thank the reviewer for the insightful review. We would like to highlight the technical novelties of our work. First, to the best of our knowledge, we are the first to formulate data poisoning defense as a forensic investigation problem, introducing a *novel problem formulation* (Section 3). While existing approaches focus on automated cleaning of suspicious samples, which requires sacrificing benign samples, we formulate defense as a budget-constrained active search with a novel objective. This perspective shifts the goal from static cleaning to a sequential, non-myopic decision process that maximizes the utility of a limited verification budget.
>
> Second, we introduce a *novel proxy objective for the search* (Section 4.2). We intuitively formulate the search objective as selecting a subset of samples to verify such that the removal of discovered poisoned samples minimizes the test loss on a clean testset. Since directly optimizing this objective is not only computationally expensive due to the need for retraining but also impossible due to the lack of test labels, we introduce a practical proxy objective for the search. We describe how we can calculate this objective efficiently in practice (Section 4.2) and demonstrate empirically that it is a good proxy for the real search objective (Section 6.3).
>
> Third, we introduce a *significant extension of Efficient Nonmyopic Search* (Jiang et al., 2017) by incorporating our proposed proxy objective and enabling search over high-dimensional image data (Sections 4.1 and 4.3). Standard active search algorithms assume uniform value across targets; we relax this assumption by assigning non-uniform values to the samples based on our proxy objective. As part of this extension, we change both the initial selection and the subsequent search steps to consider influence scores. We also derive novel upper-bounds for effectively pruning the search space, which significantly reduce the computational cost of the search (Equation (13)).

---

> > ### Author Response · Authors · 2025-11-23
> >
> > ### W2. Justification of Label-Free Influence Proxy
> >
> > The reviewer raises an interesting question: is our label-free influence metric (i.e., *influence on L1 norm of model logits*) a good proxy for the true objective of labeled test influence? To demonstrate that it is a good proxy, we followed the reviewer's suggestion and conducted additional experiments to measure the correlation between the two metrics over poisoned samples. We compared our proposed metric to two alternatives: influence on *L2 norm of model logits* and on *entropy of the class distribution output by the model*. Each metric provides a label-free quantification of the impact that a poisoned sample may have on the test loss, and therefore, each could serve as a practical proxy for the true objective.
> >
> > We conducted these experiments on the CIFAR-10 dataset poisoned using FC, BP, and GM attacks with $\epsilon = 16$. For each attack and each metric, we computed the Spearman correlation between the metric and the true objective over all poisoned samples. We have included these new results in the paper as Table 4 (see Section 6.3 in the revised paper PDF) as well as in the table below. The table shows the mean correlation ($\pm$ its standard deviation) over 20 poisoned datasets in each case.
> >
> > | Attack Type | L1 norm | L2 norm | Entropy |
> > | ----------- | ------- | ------- | ------- |
> > | FC          |**0.39 $\pm$ 0.08**|0.33 $\pm$ 0.13|0.18 $\pm$ 0.14|
> > | BP          |**0.42 $\pm$ 0.10**|0.37 $\pm$ 0.05|0.06 $\pm$ 0.08|
> > | GM          |**0.22 $\pm$ 0.25**|0.22 $\pm$ 0.26|0.09 $\pm$ 0.27|
> >
> > The results shows that our proposed, L1-norm-based influence metric achieves the highest correlation with the true objective of labeled test influence, and it does so with lower or comparable variance as the L2-norm and entropy-based metrics. This indicates that our proposed metric provides a stable and reliable proxy of the true objective, making it the best choice in practice for the forensic investigation of poisoned datasets.

---

> > > ### Author Response · Authors · 2025-11-23
> > >
> > > ### W3. Evaluation Breadth
> > >
> > > We thank the reviewer for the suggestions.
> > >
> > > To broaden the threat models considered in our experimental evaluation, we have included a new backdoor poisoning attack. We chose **Narcissus**, a state-of-the-art backdoor attack (Zeng et al., 2023b). We evaluated our proposed approach and both baseline methods, Meta-Sift and EPIC, on the CIFAR-10 dataset poisoned by Narcissus. We have included these new results in Table 1 (see revised paper PDF) as well as in the table below.
> > >
> > > |Narcissus ($\epsilon = 16$)|Undefended|Influence-Guided Active Search|Meta-Sift|EPIC|
> > > | --------------------------- | ---------- | ------------------------------ | ---- | --------- |
> > > |Attack Success Rate (↓) | 91.20% | **7.09%** | 45.54% | 11.95% |
> > > |Test Accuracy (↑) | 0.9521 | **0.9551** | 0.9526 | 0.7360 |
> > >
> > > The results demonstrate that our proposed approach is *effective against backdoor poisoning* since it achieves significantly lower ASR than the cleaning methods, while also achieving higher test accuracy. Notably, the results very clearly illustrate the advantages of a forensic investigation: automated cleaning methods face a trade-off between suffering high ASR (Meta-Sift) or low test accuracy (EPIC).
> > >
> > > We are currently running experiments with Narcissus on the *Tiny ImageNet* dataset. We will include the results in the paper as soon as they are ready.
> > >
> > > In addition, we are also running experiments with a new defense baseline, **PureGen**, a state-of-the-art data cleaning method (Pooladzandi et al., 2024). We will include the results, which compare our proposed approach to PureGen on datasets poisoned by each attack, in the paper as soon as they are ready.

---

### Meta-Review · Area_Chair_saZv · 2025-12-17

**Summary:**

Reviewers generally agreed that the paper presents a well-motivated and practically relevant reframing of data poisoning defense as a budget-constrained forensic investigation, but raised several concerns that informed the decision. The primary concern was limited technical novelty, as the method largely integrates existing ideas from influence functions and active search rather than introducing a fundamentally new algorithm. Reviewers also questioned the heuristic nature of the label-free influence proxy, initially noting insufficient justification for why the L1-logit metric reliably reflects test-time impact.

In addition, reviewers expressed concerns about the scope and breadth of the evaluation, particularly the initial focus on clean-label attacks and comparisons against a limited set of baselines. There were also questions regarding assumptions and practicality, including reliance on a verification oracle, potential dependence on poison sparsity or local consistency, hyperparameter sensitivity, and the computational cost of influence estimation.

Overall, while the approach was viewed as promising and empirically strong, these concerns about novelty, justification, and generality contributed to a borderline assessment.

**Reviewer Concerns:**

Concerns addressed by the rebuttal.
The rebuttal satisfactorily addressed several key concerns raised by the reviewers. In particular, the authors expanded the evaluation beyond clean-label attacks by adding experiments on a state-of-the-art backdoor attack (Narcissus), demonstrating that the proposed framework generalizes to different threat models. They also strengthened the justification of the label-free influence proxy by providing new correlation analyses against alternative proxies, showing that the L1-logit influence is a more reliable estimator of test-time impact. Concerns regarding hyperparameter sensitivity and failure cases (e.g., all-clean initial selections) were addressed through additional ablation studies and clarifications. Finally, the authors clarified the computational cost of the influence estimation and the role of the verification oracle in realistic forensic settings.

Concerns still outstanding.
Some concerns remain partially unresolved. While the evaluation was broadened, comparisons with several recently proposed strong baselines are still incomplete or ongoing, limiting a definitive assessment of relative performance. In addition, the core concern about limited technical novelty—namely that the approach primarily integrates existing components into a new framework—remains a matter of interpretation rather than something fully resolved by additional experiments. Finally, questions about scalability to much larger models and datasets, as well as robustness under highly adaptive or non-sparse poisoning scenarios, remain open.

**Reviewer Scores:**

Reviewer VVQc:
This reviewer already rated the paper slightly above the acceptance threshold. Given that their main concerns regarding the justification of the label-free influence proxy and evaluation breadth were directly addressed with new experiments and analyses, their score would likely remain the same or increase slightly (e.g., from 6 to a 6–7).

Reviewer yoJN:
This reviewer was generally positive but cautious, mainly concerned about hyperparameter sensitivity, initialization issues, and computational cost. Since these points were addressed through additional ablations and clarifications in the rebuttal, the reviewer would likely maintain their original score or marginally increase it (remaining around 6).

Reviewer AY6X:
This reviewer gave the lowest score, primarily due to concerns about scope, applicability beyond clean-label attacks, and missing comparisons with stronger baselines. The added backdoor (Narcissus) experiments and clarification of assumptions would likely improve their assessment. However, given that some baseline comparisons remain incomplete, their score would likely increase modestly (e.g., from 2 to 4–5), but may still remain below the acceptance threshold.

---

### Decision · Program_Chairs · 2026-01-26

Reject